# The Role of “Physiologically Based Pharmacokinetic Model (PBPK)” New Approach Methodology (NAM) in Pharmaceuticals and Environmental Chemical Risk Assessment

**DOI:** 10.3390/ijerph20043473

**Published:** 2023-02-16

**Authors:** Deepika Deepika, Vikas Kumar

**Affiliations:** 1Environmental Engineering Laboratory, Departament d’Enginyeria Quimica, Universitat Rovira i Virgili, Av. Països Catalans 26, 43007 Tarragona, Catalonia, Spain; 2Pere Virgili Health Research Institute (IISPV), Hospital Universitari Sant Joan de Reus, Universitat Rovira i Virgili, 43204 Reus, Catalonia, Spain

**Keywords:** physiologically based pharmacokinetic model (PBPK), drugs, environmental chemicals, adverse outcome pathway (AOP), machine learning

## Abstract

Physiologically Based Pharmacokinetic (PBPK) models are mechanistic tools generally employed in the pharmaceutical industry and environmental health risk assessment. These models are recognized by regulatory authorities for predicting organ concentration–time profiles, pharmacokinetics and daily intake dose of xenobiotics. The extension of PBPK models to capture sensitive populations such as pediatric, geriatric, pregnant females, fetus, etc., and diseased populations such as those with renal impairment, liver cirrhosis, etc., is a must. However, the current modelling practices and existing models are not mature enough to confidently predict the risk in these populations. A multidisciplinary collaboration between clinicians, experimental and modeler scientist is vital to improve the physiology and calculation of biochemical parameters for integrating knowledge and refining existing PBPK models. Specific PBPK covering compartments such as cerebrospinal fluid and the hippocampus are required to gain mechanistic understanding about xenobiotic disposition in these sub-parts. The PBPK model assists in building quantitative adverse outcome pathways (qAOPs) for several endpoints such as developmental neurotoxicity (DNT), hepatotoxicity and cardiotoxicity. Machine learning algorithms can predict physicochemical parameters required to develop in silico models where experimental data are unavailable. Integrating machine learning with PBPK carries the potential to revolutionize the field of drug discovery and development and environmental risk. Overall, this review tried to summarize the recent developments in the in-silico models, building of qAOPs and use of machine learning for improving existing models, along with a regulatory perspective. This review can act as a guide for toxicologists who wish to build their careers in kinetic modeling.

## 1. Introduction

PBPK models are mathematical models encompassing multiple compartments with physiology, anatomy, biochemical and physicochemical parameters for describing ADME (absorption, distribution, metabolism and excretion) of xenobiotics and their metabolites (Figure 1) [1,2,3,4]. These models vary from empirical, semi-mechanistic to compartmental models based on the complexity of the problem [5]. The major challenge with empirical or semi-mechanistic models is their difficulty with interpreting questions such as “How to predict concentration–time profile of compound in target organ” or “How to accurately predict the exact dosing while extrapolating from animal to human”. This led to the development of compartmental models, which are close to human anatomy and physiology. Currently, these models are widely acknowledged in the field of pharmaceutical and environmental science for the prediction of PK behavior of xenobiotics (drug/chemical) with respect to dose, route and species [6].

In the past, allometric scaling was used to predict some PK parameters such as clearance and volume of distribution for humans from PK profiles in animals or other species, as well as for sensitive populations such as pediatric, but this scaling fails to predict the parameters in diseased or compromised states. In addition, it is a bit challenging to incorporate the drug–drug interaction or mixture assessments during allometric scaling, which has led to the reduction in popularity of such approaches [4,5,7]. PBPK models can overcome such challenges, as they consist of systems data (such as physiology) and biochemical parameters such as metabolism and excretion, including mechanisms such as saturation of enzymes or presence of specific receptors [8].

PBPK models have multiple applications, and, recently, toxicokinetics is being integrated with adverse outcome pathways (AOPs) for improving overall risk assessment in the context of new approach methodology (NAM) [9]. In-vitro systems are currently accepted in drug discovery and toxicological assessments, especially for predicting molecular-initiating events (MIEs) and key events (KEs) [10]. However, the question arises if the chemical is reaching a high enough concentration in target tissue to cause a perturbation, which can affect cellular biology and cause adverse effects. This points towards quantification of kinetics for evaluating the internal concentration at the MIE site considering all the possible scenarios during in-vitro testing such as binding of chemical to lipids and proteins, partitioning between medium and chemical and binding to the plastic in the cell culture system, which can reduce the free chemical. IVIVE-based mechanistic PBPK models (integral components of NAM) can help in taking all the factors into consideration for predicting the kinetic and also the in-vitro exposure equivalent to the human target organ for successful human health risk assessment [11,12]. The development of PBPK models is sometimes considered time- and resource-intensive and complicated due to multiple parameters, with many having unknown values, which are often fitted. Therefore, utilizing machine learning and artificial intelligence approaches can help in predicting some of the PK parameters of new substances, which can speed up the process of developing robust NAM-based PBPK models with limited experimental data [13,14].

In this review article, we summarize the methods to develop a PBPK model and feasibility to extend them for different human populations and diseased patients. The focus is also on existing challenges in PBPK modeling and overcoming them by building organ-specific models and on their application for AOP development. The comprehensive idea about the usage of machine learning and artificial intelligence for improving and advancing the existing PBPK framework is also discussed. Additionally, the literature gap and ways by which regulatory acceptance of these models can be improved is mentioned, providing a complete picture of current PBPK models.

## 2. Unravelling the Art of Developing PBPK Models

The PBPK model explicitly considers different organs, which are called compartments, but the choice of the number of compartments solely depends on the creator and required outcome. Typically, the compartments utilized are the liver, gut, kidney, heart, stomach, spleen, brain, muscle, bone and skin. However, some researchers prefer two-compartmental models, which are easy to build (Figure 2) [6]. They depend on the assumption that the compound is being absorbed into the gut and then moving to the primary compartment through excretion in urine (Equations (1)–(3)) or that the compound is moving from the central compartment to the peripheral compartment, followed by excretion. The challenge being that these models fail to capture the complex human physiology and events such as metabolism, reabsorption etc.

The simplistic equation for a two-compartment model is:(1)$$\frac{d(A_{gut})}{dt}=-k_{abs}*A_{gut}+oral\;dose$$
(2)$$\frac{d(A_{pri})}{dt}=k_{abs}*A_{gut}-k_{el}*A_{pri}$$
(3)$$\frac{d(A_{urine})}{dt}=k_{el}*A_{pri}$$

The differential equation (*dA*/*dt*) defines the amount of compound with time (*gut*, primary compartment, or *urine*), *k_abs_* refers to the absorption rate constant and *k_el_* is the elimination rate constant. All the amounts shown in equations can be converted to the concentration by considering apparent volume of distribution for a two-compartment model, whereas, for a multi-compartment model, this is done by dividing the amount with the volume of a particular tissue.

Multi-compartment models often provide detailed information about various tissues and, most importantly, they include intrinsic physiological processes such as liver metabolism, gut metabolism, kidney transporters and elimination, reabsorption and enterohepatic recirculation (EHR) to capture the linear and non-linear kinetics along with anatomy (Figure 3) [3,4,15,16,17]. However, a point worth mentioning is that adding a lot of compartments often makes the model complex and impractical, since values are required for multiple parameters. Often, the choice of compartments is governed by target chemical, species and the route of administration.

### 2.1. Model Parameterization

#### 2.1.1. Physiological Information

For building any PBPK model, we need anatomical and physiological parameters of the respective species, such as rat, dog, human, etc. [5,9]. Parameters include body weight, height, organ volume, cardiac output and organ blood flow. There is some literature available where authors have collected the physiological data or developed equations for human lifetime [3,18]. Mass balance equations are essential to explain total blood flows and other parameters connecting all tissues. All PBPK models contain a compartment called “rest of the body”, which can be calculated for volume by subtracting the body weight from total volume of other organ and for blood flow by subtracting the cardiac output from total blood flow for other organs, respectively. These data can be fed into the PBPK model to account for variation in the ADME profile based on the physiology.

#### 2.1.2. Biochemical Information

There are multiple biochemical parameters, but the most basic are (a) absorption and EHR, (b) distribution and fu, (c) metabolism with IVIVE (in-vitro to in-vivo extrapolation) and (d) excretion. These parameters can be either calculated based on PK principles or fitted using multiple methods such as Bayesian with a Markov Chain Monte Carlo (MCMC) algorithm, Metropolis–Hasting algorithm, reversible jump MCMC, the Gibbs sampler and Hamiltonian Monte Carlo (HMC) [19]. The choice of algorithm depends on the ease of use and type of clinical data available.

##### Absorption and Enterohepatic Recirculation

Absorption is defined as the movement of a xenobiotic from its site of delivery to systemic circulation or another compartment [20]. In the PBPK model, absorption is defined by the absorption rate constant (*k_abs_*), which is the rate of absorption of the xenobiotic in the system. It is often a first-order rate constant. This parameter becomes tricky when, instead of a simple formulation, there is a delayed absorption, which can be due to a modified formulation or a meal effect [21]. Parameters such as intestinal permeability or dissolution rate can be measured to calculate the *k_abs_*. Absorption-related processes such as EHR need to be introduced sometimes to predict the terminal phase of the plasma-concentration time profile (Figure 4) [4,22,23]. Compounds that have high biliary excretion, faster absorption rates and limited clearance are the substrates for EHR [24]. The equations for EHR often include liver, bile and gut as the main compartments (Equations (4) and (5)).
(4)$$\frac{d(A_{liver_{1}})}{dt}=k_{abs}*A_{gut}+Q_{liver}*C_{x}-k_{liver-bile}*A_{liver1}*C_{liver1}$$
(5)$$\frac{d(A_{bile})}{dt}=k_{liver-bile}*A_{liver1}-k_{bile-gut}*A_{bile}$$

In Equation (4), *k_abs_* is the absorption rate constant, *A_gut_* is the amount of compound in the gut (mg), *Q_liver_* is the blood flow to the liver, *k_liver-bile_* is the rate constant for the transfer from liver to bile and *A_liver_* is the amount of chemical in the liver (mg). In Equation (5), *A_bile_* refers to the amount in bile (mg), *k_bile-gut_* is the bile gut rate constant changing over time, representing transfer from bile to gut, and *A_liver_* is the amount in the liver (mg). The bile–gut transition is a first-order rate constant, and it could change with time based on bile flow rate, which can be described by a turnover model. More details about this model can be found in this article [25]. This phenomenon become important in the case of non-linear kinetics.

Sometimes, the experimental data show multiple peaks and the generic PBPK framework fails to capture it. For instance, a PBPK model was developed for DPHP using lymphatic uptake and EHR to predict the plasma and urine concentration, including the delayed peak. The author explained that the inclusion of three processes (systemic circulation to lymph movement, high protein binding and efflux of absorbed chemical from liver via hepatic route) were able to explain the data [26]. The important point here is that, for kinetic modeling, it is imperative to understand the experimental data and gain mechanistic knowledge before diving into the calibration and optimization of the model.

##### Distribution and Fraction Unbound

Fraction unbound is of the utmost important input parameters, which can affect the output from the PBPK model. Albumin, alpha acid glycoproteins (AAG) and lipoproteins (LPs) are among 60 plasma proteins that bind to drugs/chemicals [27]. Basic compounds bind to AAG and LPs, while acidic and neutral compounds bind to albumin. Generally, for environmental chemicals, the experimental fup for adults as well as children is not available [27]. For adults, in silico models such as QSPR, based on the learned relationship between structure and protein binding, can help. For fup in infants, equations are available in the literature [28] for extrapolating fu from adults (Equation (6)). Ye et al. demonstrated that a wrong calculation of fup could result in Vss prediction error for neutral drugs and error in clearance for low-cleared compounds [29], pointing towards the importance of fu in PK.
(6)$${fu}_{infant}=\frac{1}{1+\frac{P_{infant}}{P_{adult}}\frac{\left(1-{fu}_{adult}\right)}{{fu}_{adult}}}$$

Here, *P* refers to molar protein concentration.

Another important parameter in PBPK models is the partition coefficient for a perfusion-limited model, required to distribute the xenobiotic in tissues. The partition coefficient for PBPK can be computed either by calculating AUC based on experimental data (Equation (7)) or by the equation derived by multiple authors [30,31,32,33,34]. Among these, the Rodger approach is widely accepted for most drugs due to consideration of binding with lipids and proteins. Nonetheless, all these methods need tissue composition data and physiochemical properties as inputs to quantify the organ: plasma/blood partition. In a recent study, a standardized tissue composition was proposed for humans, which can be used as a common input for five partition coefficient prediction methods [30,32,33,35,36]. The author compared the results utilizing a PBPK model and found that all partition coefficient methods should be considered in the process of drug development [37].
(7)$$Partition\;coefficient=\frac{{AUC}_{organ}}{{AUC}_{plasma/blood}}$$

##### Metabolism with IVIVE

Metabolism through hepatic and extrahepatic enzymes is an important aspect from the PK perspective and account for toxicity or detoxification. Most parent drugs and chemicals are metabolized by phase I and phase II reactions into several metabolites or inactive excretion products in the liver. Phase I reactions are majorly catalyzed by cytochrome P-450 (CYP) enzymes, and phase II by UDP-glucuronosyltransferase (UGT-s), sulfotransferase (SULT), glutathione S-transferase (GSTs), N-acetyl transferases (NAT), etc. [38]. Apart from this, some drug transporters such as ATP binding cassettes (ABCs) and solute carrier transporters (SLCs) are responsible for influx and efflux of compounds and metabolites contributing to phases 0 (absorption) and III (elimination) [39]. All these metabolic kinetics can be incorporated in PBPK model with specific parameters such as enzymatic expression in organs, maximum rate of metabolism (Vmax) and Michaelis constant (Km), especially for non-linear metabolism (Equation (8) [40].
(8)$$\frac{dp}{dt}=\frac{V_{max}\cdot \left[S\right]}{K_{m}+\left[S\right]}$$
where *p* is the product and *S* is the substrate.

Vmax can be calculated in the model using the IVIVE technique, utilizing in-vitro data with the following Equation (9). Vmax is in pmol/min/pmol of *enz*, *enz* refers to enzyme abundance in a particular organ such as the liver (pmol/mg of microsomal protein), MPPGL is microsomal protein per gram of human liver (mg/g), *V_org_* refers to organ volume (L) and *MW* is molecular weight of the chemical (g/mol) [17,41]. Since *Vmax* is in mg/h, 60 is used for the conversion from min to h, and 10^6^ refers to the unit conversion (mg). This equation can be incorporated in the PBPK model to account for metabolism in the gut, intestine or any other organ. Additionally, metabolic kinetic data for the liver are often presented in the form of Vmax (in-vitro/tissue), conducted using either hepatocytes, microsomes, cytosol or whole organ culture. IVIVE scaling requires extrapolation, which can be done based on cell type, hepatocytes: 74–131*10^6^ hepatocytes per gram of human liver, or 34 mg/g microsomal protein per gram of liver for microsomes [42].
(9)$$V_{max}\left(\frac{mg}{h}\right)=V_{\max\_\mathrm{pmol}(\mathrm{in}-\mathrm{vitro})}*enz*MPPGL*V_{org}*MW*60/10^{6}$$

##### Excretion

Excretion can mainly happen through urine, feces and via exhalation, depending on the compound. Other routes of excretion may include lactational, saliva, mucosal, sweat, etc., but, generally, they account for a minor fraction and, hence, are not considered. However, based on the toxicological risk, they can be included in the PBPK model.

In one–two compartment models, excretion via urine can be through plasma, explained by Equation (3) (mentioned earlier). Even in multi-compartment models, many authors modeled urinary excretion of compound directly from plasma or blood, respectively [15,43]. The excretion from the kidney is also modeled mostly by first-order kinetics, which is the simplest expression in first-order excretion (Equation (10). Here, *k_el_* refers to elimination rate constant and *Akidney* is amount of compound present in the kidney.
(10)$$Urine=k_{el}*Akidney$$

For long-acting compounds such as PFOA, multiple mechanisms such as glomerular filtration rate (GFR), basolateral active transport and apical transport in proximal tubular cells, along with simple diffusion, are often modeled to account for slow renal excretion [44]. In one specific study, a generic PBPK model was developed, including mechanistic kidney (proximal, loop of Henle, distal and collecting segment, Bowman’s capsule, GFR and bladder), incorporating active secretion and bidirectional passive diffusion. The model was able to predict very well the renal clearance for 46 drugs (87% values within two-fold) utilizing in-vitro permeability data [45]. This shows the application of PBPK in utilizing in-vitro data for parameterization of the model.

Excretion by feces is possible if the compound excretes in bile or excretes through intestine or is not absorbed in the GI tract [22]. In PBPK, simply excretion can be modeled using first-order kinetics with Equation (11).
(11)$$Feces\;=K_{el}\cdot Afeces$$

Very few compounds are being excreted via exhalation, mostly being volatile organic compounds. The PBPK model incorporates a tissue and air partition coefficient (*P_b_*) to account for respiratory equilibrium (Equations (12)–(14)). The arterial blood is dependent on cardiac output (*Qc*), concentration in venous blood (*C_v_*), alveolar blood flow (*Q_alv_*) and *C_inh_* (inhaled concentration). Exhaled concentration can be provided by Equation (14), assuming 70% alveolar respiration [46]. However, the inclusion of this route is quite limited for PBPK in pharmaceutical drugs and chemicals due to most compounds being non-volatile in nature.
(12)$$C_{a}=\frac{\left(Q_{c}\cdot C_{v}\;+\;Q_{alv}\cdot C_{inh}\right)}{\left(Q_{c}\;+\;\frac{Q_{alv}}{P_{b}}\right)}$$
(13)$$C_{alv}=\frac{C_{a}}{P_{b}}$$
(14)$$C_{exh}=0.7\cdot C_{alv}\;+\;0.3\cdot C_{inh}$$

### 2.2. Route of Administration

The PBPK model holds the flexibility of adding one or multiple routes based on the exposure. The oral route is the most common, both in drugs and environmental chemicals, which often limits the bioavailability due to first-pass metabolism. Nonetheless, oral is most preferred due to ease of administration and considered as the alternative to intravenous (IV) infusion depending on the specific case. For instance, a gemcitabine PBPK model was developed using Gastroplus software with IV and oral routes of administration to evaluate the plasma-concentration time profile. The author showed that the drug Cmax was lower, but AUC (area under curve) was higher for the tablet dosage regimens (1000 mg, three/day and 1500 mg 2–3 times a day) compared to the IV infusion [47]. Often, the availability of the compound in systemic circulation is dependent on multiple factors. For instance, absorption is often considered the critical parameter for the oral route along with influence from fasting or fed state.

Compounds such as bisphenols, which can be absorbed through skin and have higher exposure for certain people (e.g., cashiers), require specific dermal PBPK models [48]. It is observed that dermal exposure of this compound results in higher half-life (approx. 8 h) due to bypassing first-pass metabolism [49,50]. A pregnancy PBPK model was developed for BPA including both oral and dermal exposure [17]. Another PBPK model was developed for BPA and its analogues considering the dermal exposure. Dermal exposure was found to be the predominant route along with peroral, and BPS exposure led to the highest internal concentration of unconjugated compound [51].

The inhalation route is quite successful in respiratory diseases and also for rapid systemic drug delivery. A PBPK model was built for inhaled nemirasalib, consisting of extra-thoracic, thoracic, bronchiolar and alveolar tissues for evaluating pulmonary drug absorption [47]. For environmental chemical such as xylene, PBPK with lungs as a route for inhalation and exhalation was modeled for reconstructing the human exposure [52]. Another model was for ethanol inhalation through first-generation and second-generation electronic cigarettes [53]. The study found that estimated BAC results were below the toxicological threshold, showing the application of screening exposure assessment for safety of the product.

For some compounds such as chloroform, where ingestion and inhalation is the most significant route, multi-route PBPK models are required. The author showed utilization of a Bayesian approach in PBPK, considering ingestion, inhalation and dermal routes for typical household exposure scenarios. Realistically, for environmental chemicals, multi-exposure PBPK models are required, but, since the exposure from some routes is minor, it is often ignored. However, in the case of drugs, mostly the exposure is defined, so multi-route modeling is not required.

### 2.3. Model Simulation

The last step in building a PBPK model is to introduce all the parametric values in the software or write your own code for simulating the output. Some of the software for developing PBPK models are GastroPlus, PK-Sim, BioDMET, Maxsim2, PKQuest, Phoenix WinNonlin and Berkeley Madonna, which are very specific and easy to use. However, most of them are commercial and, hence, not freely available. Free, available and generalized software include GNU MCSIM, COPASI, PKSim, etc., which also have easy interfaces. Additionally, users can write their own code in software such as R and python, which utilize solvers such as desolve and LSODA for ODE solving. In Rstudio, there are specific packages such as httk (high throughput toxicokinetic) dedicated to make the simulation easy by containing chemical-specific data and physiological information [54].

After building the model, the next ideal step is to validate it with experimental data before moving towards prospective predictions. Generally, the simulated and predicted result within two-fold is considered as good validation for building robust PBPK models, but it is mostly applicable for wide therapeutic indexed drugs and chemicals. Sometimes, it is not possible to validate the output from all compartments of the model. In these scenarios, the robustness of the model is ensured based on well-supported assumptions with high confidence in underlying mechanisms [55].

## 3. Applications of PBPK Models for Pharmaceuticals and Environmental Risk Assessment

### 3.1. PBPK Model Capturing Sensitive and Diseased Populations

Originally, the PBPK models were focused on exposure prediction and toxicokinetics of xenobiotics for adult populations [56]. Over time, they have matured and, currently, are being utilized for sensitive populations such as infants and patients suffering from renal or kidney diseases (Figure 5). The study by Michelet et al. utilized propofol for developing PBPK models in children and neonates (term and preterm) using top-down and bottom-up approaches [57]. Propofol is extensively metabolized in the liver and kidney, with glucuronidation being the major pathway. An adult model was extrapolated to the pediatric population, incorporating changes in tissue volume, blood flow and variation in ontogeny function with age. The model was able to predict the concentration–time profile similar to experimental data for preterm neonatal population. Similarly as for BPA, a pediatric PBPK model was developed after extrapolating from adults, considering ontogeny changes to account for age- and gender-specific risk [4].

Different software have different physiological information, ontogeny functions for enzymes and principles for utilizing unbound drug concentration in hepatocytes for pediatric populations. For instance, the Simcyp model uses unbound drug concentration with unionized fraction as the major driving force for metabolism, whereas Gastroplus utilizes unbound with both ionized and unionized concentration for metabolism [58]. In PK-Sim, drug–tissue distribution is being handled by permeability*surface area product and tissue–plasma partition coefficient. In this software, based on step speed, perfusion or permeability limited model runs for the output. Such differences lead to significant variation in PK parameters in pediatric populations. Additionally, the physiological parameters used to build these models were taken from different datasets and literature sources and, hence, there is variation in the values. There is a need to harmonize the pediatric dataset and extrapolation principle for improving the existing PBPK models. In addition, besides refining these models for small molecules, they should be also extended to therapeutic proteins and large molecular entities to evaluate age-related changes that are impossible to predict in-vitro or in-vivo [59].

Another application of PBPK is predicting compound concentration in geriatric populations, which is the largest population for the pharmaceutical market. Unfortunately, elderly populations aged over 65 have the least studies in clinical trials, but they are the biggest consumers, taking two to five types of medication per day [60]. A geriatric PBPK model was modified for the Chinese population, incorporating physiological parameters and drug-dependent parameters for six drugs. Refinement of the model based on age, weight, height, BSA, creatinine levels, etc., improved the prediction performance of PBPK, assessing age-specific risk. A human lifetime PBPK model was developed for forever chemicals such as PFOS to evaluate long-term risk and disposition in organs for improving risk assessment [3].

By incorporating pathophysiological changes occurring in a disease, PBPK models are being extended for drug-disease models. They can be used for predicting the ADME of a given drug or chemical in the case of liver or renal failure. For instance, Rasool et al. developed a PBPK drug-disease model for rifampicin in tuberculosis and cirrhosis patients [61]. Variation in albumin concentration in TB patients improved the clearance prediction. Simulated results showed an increase in AUC in cirrhosis patients after oral dosing. Such a model can be helpful in dose selection for diseased patients. PBPK models can provide valuable clinical insight about dosage design in diseased patients such as those with renal failure, but, still, they are not being widely accepted by regulatory bodies such as the U.S. Food and Drug Administration (FDA). However, PBPK model applications for renal incidences have been highlighted in the FDA guidance of 2020 for supporting inclusion of patients with RI in clinical trials for improving the dosage regimen [62]. Kidney disease affects the PK profile of both renally and non-renally cleared drugs impacting the likelihood of drug- or chemical-induced toxicity. In such cases, specialized PBPK models such as organ-specific ones can unravel the true kinetics or the mechanism happening inside specific sub-compartments.

### 3.2. Current Trend towards an Organ-Specific PBPK Model

Organ-specific PBPK models are next-generation kinetic models that are one-step ahead in evaluating the kinetics inside a specific organ (Figure 5). Multiple organ-specific models with focus on liver, heart, brain, kidney, etc., have been developed by researchers. Inclusion of permeability-limited models with perfusion-limited PBPK improves prediction of transporter-mediated interaction for specific organs. Often, the integration of these models is being used by researchers to define the complex PK of some drugs or environmental chemicals [63].

Translational models for antibody disposition in the brain consist of the blood–brain barrier (BBB) and blood–cerebrospinal fluid (CSF) barrier, with two separate sub-compartments: the CSF circulation system and brain parenchyma [64]. This model predicts quantitatively monoclonal antibodies’ PK at the targeted site in rats, mice, monkeys and human and, hence, can be used for preclinical to clinical translation for CNS disorders. Another population-PBPK model was developed based on brain micro dialysis data, and a Simcyp model was used for bottom-up prediction to model CNS with brain ECF and brain tissue [65]. Interestingly, both the models provided similar predictions in the brain, but ECF concentration was accurately predicted by population-PBPK. In short, the type of PBPK model to be utilized depends on the output required and mechanistic knowledge regarding the ADME profile of drug along with in-vitro [66] and in-vivo data. PBPK with sub-compartments such as globus pallidus, pituitary gland, olfactory bulb, cerebellum and blood brain have been developed for compounds such as manganese to evaluate the CNS toxicity [63]. These are multiple examples of successful PBPK models developed either to capture the kinetics in the hippocampus, cortex or ECF section of the brain [65,67] to target neurological disorders or neurotoxic risk assessment.

For cardiac toxicity and safety assessment, PBPK models consisting of four heart compartments, epicardium, midmyocardium, endocardium and pericardial fluid, were developed in R software. A whole-body PBPK model of seventeen compartments was built and integrated with a heart model along with cardiac metabolism using CYP450. The heart model was used to predict amitriptyline concentration in plasma and heart and was a first attempt to develop an organ-specific model for this case scenario [41].

Forever chemicals such as PFOA, which are not metabolized and renally reabsorbed, need a specialized PBPK model accounting for transporter-mediated renal elimination. A PBPK model involving kidney transporters was developed for both sexes of rats, incorporating proximal tubule lumen, proximal tubule cells with diffusion and active transport to defined PFOA kinetics. Activity in proximal tubule cells and kidney transporters were found to be important components for PFOA serum clearance and also the half-life for both sexes [68]. Recently, the mechanistic kidney model incorporating ontogeny and physiology of renal processes was able to predict the PK of five renally excreted drugs for pediatric populations [69]. A permeability-limited model was incorporated with the conventional model for evaluating the effect of cimetidine on the kinetics of metformin [70]. PBPK incorporated active uptake and efflux with an electrochemical driving force for OCT1 (organic cation transporter) and OCT2 in the kidney and permeability limited uptake in the liver with mechanistic modeling to define PK. The model was able to predict OCT and MATE-mediated (MATE: Multidrug and toxin extrusion) drug–drug interactions, highlighting the importance of such models in drug development. A mechanistic kidney model was integrated with full-body PBPK to evaluate the urine pH effect on the drug ADME profiles for methamphetamine and amphetamine [71]. The model was successfully able to simulate the plasma-concn. time profile and urinary profile for both compounds. Such kind of models can evaluate the effect of urine pH on drug renal excretion, avoiding the toxicity, in the case, of patients who are taking multiple medicines. In a nutshell, such organ-specific models have the potential for predicting toxicity and improving risk assessment for sensitive populations.

### 3.3. PBPK Role in IATA and AOP

Integrated approaches for testing and assessment (IATA) focus on chemical safety, minimizing environmental and human health risk utilizing advanced in-silico, in-vitro and in-chemico approaches. In IATA strategy, AOP is one of the frameworks helpful in developing it through evaluating relationships between key events (KEs) and adverse effects and identifying data gaps (Figure 5 and Figure 6) [72,73,74]. Mostly, the developed AOPs are qualitative for evaluating the toxicity, but only a few are quantitative, with dose response relationships or mechanistic modeling to evaluate risk. Integrating PBPK with AOP can help in predicting the kinetics of chemicals and, hence, endpoint toxicity, utilizing the IATA approach [9]. Fusion of the PBPK and pharmacodynamic (PD) model for developing qAOPs can be one of the strategies for next-generation risk assessment [75]. AOP development along with ADME prediction can help in using AOPs for chemical-specific exposure and PK considerations for both data-rich and data-poor chemicals. For instance, PBPK was used to evaluate pollutant concentration in an organ with time for long-term exposure. It is important for risk assessment, as it may be possible that, at low exposure, the internal dose never reaches the threshold activation and, hence, may not initiate the particular event or, in this case, the chronic inflammation, as suggested by the author [76]. The other point worth mentioning is that some pollutants may not behave linearly and there can be increased accumulation due to saturated clearance. All these scenarios can be captured by detailed PBPK models providing improved estimates of cell concentration and, hence, probability of developing a particular adverse outcome (AO).

The important point here is that uncertainty and variability can be introduced in PBPK models, which may be important for qAOPs. In PBPK, we can develop a virtual population with variable physiological parameters and evaluate the toxicokinetic profile. In addition, sensitive populations and diseased populations can be considered to evaluate the activation of MIE or KE or the specific AO, which may not be possible in the in-vitro studies. Another important aspect is that sometimes in-vitro kinetics may differ from in-vivo, which is quite challenging to address. For instance, metabolic kinetics can vary in-vitro to in-vivo drastically, hence, affecting the parent compound and also activation of specific MIE (e.g., production of ROS, cellular injury) [77]. These are some of the challenges that need to be addressed. Nonetheless, kinetic models such as PBPK are helping in addressing some of the challenges through QIVIVE for building qAOPs.

In a case study, a life stage PBPK model was integrated with AOPs to describe the fetal blood levels equivalent to the concentration at which in-vitro activity related to angiogenesis/vasculogenesis was observed [72]. In-silico models such as systems biology (SB) can be integrated with PBPK to provide a broad understanding about AOPs. For instance, when developing qAOP for oxidative stress-induced chronic kidney diseases, three approaches were used: (1) dose-response, (2) Bayesian network (BN) and (3) SB model. The author found that the dose-response model provides a good fit, but they should be accompanied by the SB model to provide mechanistic quantitative understanding [78]. BN was more precise than dose-response but simpler than SB, but their usage needs more experience. In the case of quantifying the AO for humans of different ages, the PBPK model can serve as a gold standard tool to evaluate the risk.

## 4. Machine Learning with PBPK

Currently, PBPK models consist of mathematical equations that are basically ordinary differential equations (ODEs) being solved by an ODE solver to compute the output. Recent advancements in machine learning and artificial intelligence can help in advancing this field. A neural PK/PD model was developed through deep learning for patient response time course to analyze drug concentration and platelet response from 600 patients [79]. Interestingly, the neural model performed better than the pop PK/PD model, making it a promising methodology for forecasting precisely a patient’s future response [80]. Another neural ODE model developed for trastuzumab emtansine showed good performance while predicting PK for a new treatment regimen. A variety of machine learning models such as LSTM (Long short-term memory), NLME (nonlinear mixed effects), LightGBM and neural-ODE were used but are only limited to neural-ODE generated continuous PK profiles with correct concentration-time patterns in artificially simulated experiments.

Apart from training for ODE, machine learning can also be used for calculating the biochemical parameters required to build PBPK models. A machine learning algorithm was used for calculating absorption rate constant, hepatic intrinsic clearance and volume of systemic circulation using 14–26 physicochemical properties generated by cheminformatics software. A PBPK model with in-silico-derived parameters was able to accurately predict the concentration–time for plasma after oral dosing [81]. In another work, a read across and QSAR model was developed for 1487 environmental chemicals of TK parameters to estimate steady state concentration and derive bioactivity exposure ratio (BER) for helping in risk-based chemical prioritization [82]. Such machine learning techniques carry the potential to improve existing PBPK models and made them more robust for regulatory use by reducing their dependency on experimental data.

## 5. Challenges and Guidelines for Regulatory Acceptance

Advancement in PBPK and enough data for validation and calibration of the model have led to acceptance by regulatory authorities. PBPK models have improved model-informed drug development by assessing drug–drug interaction, dosing prediction, dose validation in pediatric patients, pregnancy dose calibration, lactational exposure and much more in the pharmaceutical field. The percentage of new drug approvals with PBPK analysis has increased over the years, with higher acceptance rate by the US FDA and EMA (European Medicinal Agency). A lot of workshops are being organized to inform companies and scientific personnel about advantages of using such mechanistic models and building robust models with good practices [83].

Environmental organizations such as the EFSA (European Food Safety Authority), US EPA (Environmental Protection Agency) and ATSDR (agency for toxic substances and disease registry) consider the PBPK model for risk assessment based on the strength and robustness of the model. PBPK for perchlorate was assessed by EPA based on a six-step framework and was found to be suitable for calculating human health risk for different life stages [84]. The framework can be referred to through this reference [85,86]. Recent total weekly intake (TWI) set by EFSA for four perfluoroalkyl substance (PFAS) is based on the evaluation using PBPK modeling. The lowest BMDL_10_ found in children of 1 year was used to estimate maternal exposure of 0.63 ng/Kg BW/day. Based on this value, and considering accumulation, a TWI of 4.4 ng/Kg BW/week was established [87].

PBPK models offers multiple advantages from dose selection/daily exposure prediction, drug–drug interactions, concentration–time profiles in multiple organs, etc. However, their use in assessment of trial design, pediatric formulation and toxicology are still limited [56]. There exists a scope of improving existing PBPK models for integrating them in the regulatory framework. One of the biggest challenges in PBPK model building is the estimation of unknown parameter values. Sometimes, the model parameter is assumed equivalent to the other species for the same chemical, which is acceptable. In other scenarios, the parameter value needs to be estimated based on MCMC optimization algorithms. The user should set a setting or initial values for estimation and also needs to check for convergence after the run. This is the most important step in PBPK and is not discussed in detail while developing PBPK, often reducing the confidence in the model.

The literature analysis suggests that existing PBPK models have some other gaps such as uncertainty in abundance of phase I and phase II enzymes for sensitive populations and limited knowledge about IVIVE and PBPK for non-CYP enzymes such as UGTs, SULTs, AO (aldehyde oxidase) and CESs (carboxylesterases) [83]. Lack of abundant information about polymorphism for UGTs, extrahepatic elimination and transporter-mediated chemical–chemical interactions are some gaps in PBPK models where research can be focused. Still, data about many transporters, especially abundance and activity, are missing, and limited IVIVE techniques often lead to under-confidence in the model [83,88]. Efforts are ongoing to improve the in-vitro testing and data analysis to fill some of these gaps for building robust PBPK models.

To improve PBPK, several regulatory agencies such as the EMA and FDA published guidelines on reporting of modeling and simulation to enable assessment by regulators [89,90]. For instance, all the model parameters including assumptions, system-dependent parameters, compound-specific parameters and models should be presented clearly and justified whenever required. An analysis plan should be built along with sensitivity analysis both during development and application. Details of the mathematical equations should be presented to evaluate the correctness of the structure. The most important, being mass-balance and blood-flow balance, should be supported with no numerical errors. The important parameters such as Cmax, tmax, half-life and AUC need to be reported and discussed in detail [89]. Both for academics and industry, it is important to follow these guidelines for increasing the confidence in NAM-based modeling approaches.

The PBPK model supports NAM-based research by focusing on insilico toxicology combining in-vitro testing to assess health risk. NAM has been considered by regulatory bodies for almost fifteen years, but, still, assessing biokinetic/toxicokinetic through this methodology is a bottleneck in chemical risk assessment [91]. For increasing the acceptance of toxicokinetic models such as PBPK by regulatory bodies, proper guidance protocols are a critical step along with capturing transporter-mediated processes of a particular organ. Another important point being that PBPK model development and validation lack consistency in quality assessment practices, making it difficult to reproduce the model [92]. Stringent criteria for accepting the PBPK model based on the toxic potency of the chemical or therapeutic index of the drug need to be defined. Recommendations for accepting the PBPK models need to be clearer, and criterion, such as the Akaike information criterion, and correlation analysis should be used to predict the best fit, rather than a visual approach [92] (Sager et al., 2015).

## 6. Conclusions

In this paper, we showed how to develop a single or multi-compartment PBPK model based on the requirements and provided data. These models have multiple applications in clinical, pharmacological and toxicological scenarios for both drugs and environmental chemicals. PBPK models have wide applications for diseased populations and for building qAOPs. Machine learning can help in easing the process of building PBPK through providing biochemical parameters. Currently, PBPK models are being accepted by regulatory authorities, especially for sensitive populations. Further, they can be combined with PD models to evaluate efficacy and potency of xenobiotics. Collaboration between companies, academia and health authorities is required to improve the existing models and enhance their use in predicting PK, dose/daily exposure and much more.

## Figures and Tables

**Figure 1 ijerph-20-03473-f001:**
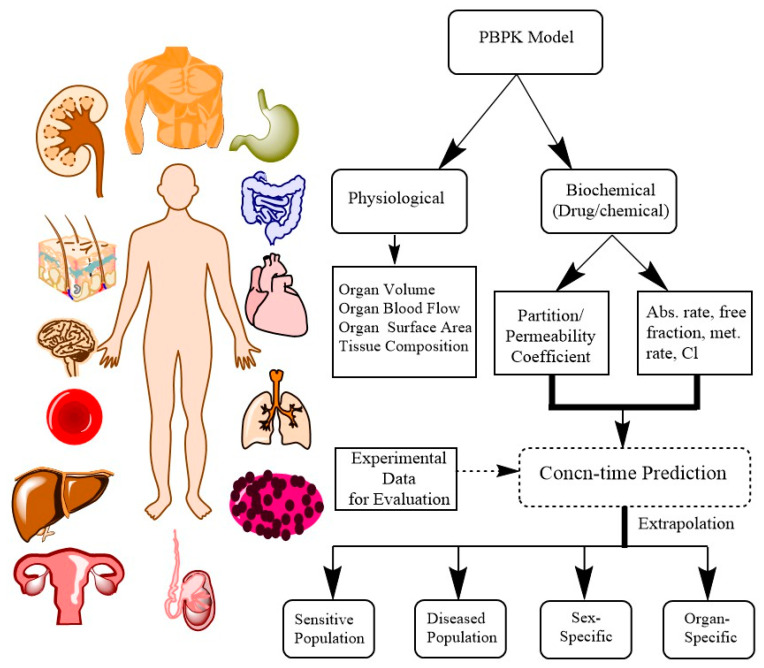
Conceptual diagram for PBPK model representing physiological and biochemical parameters and extrapolation of the model for human health risk assessment.

**Figure 2 ijerph-20-03473-f002:**
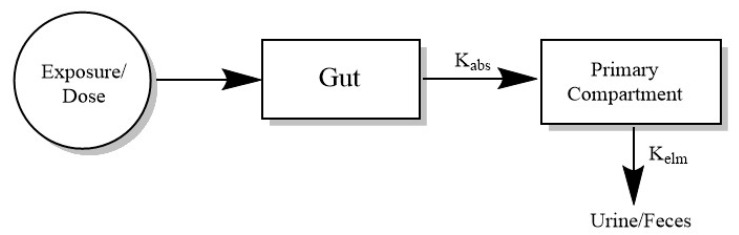
Overall structure for building a two-compartment model considering absorption and elimination rate constant.

**Figure 3 ijerph-20-03473-f003:**
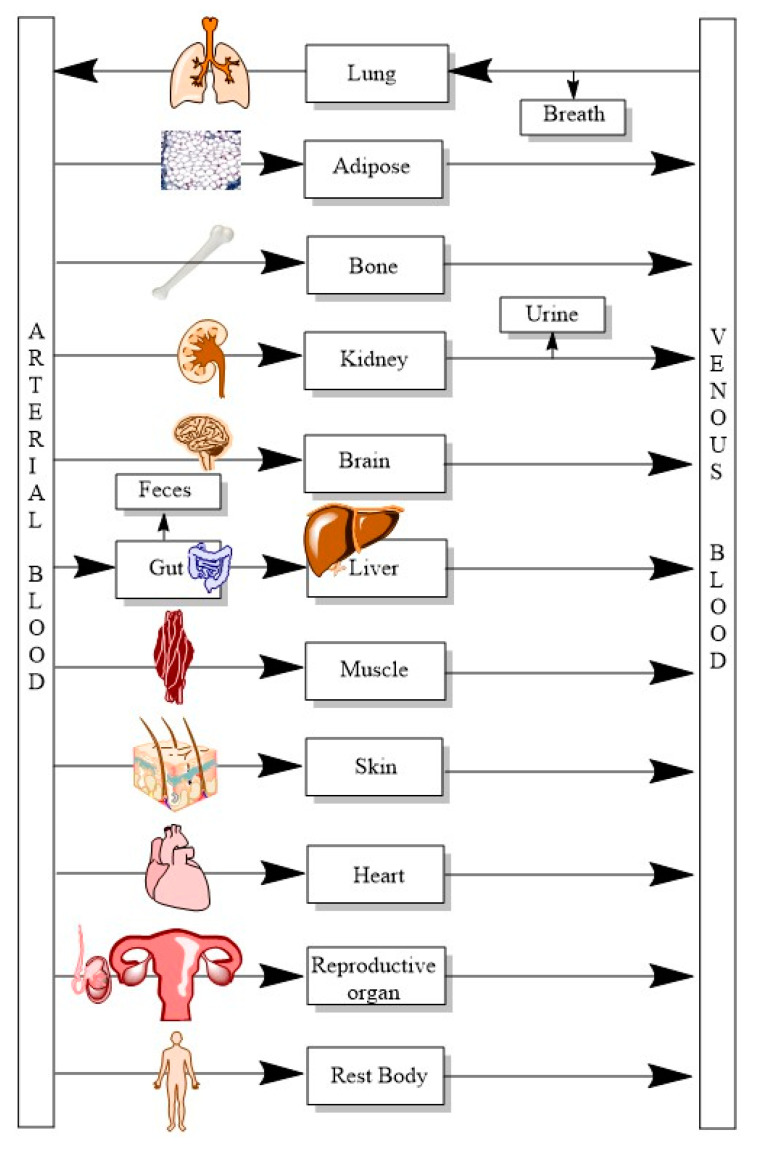
Multi-compartment PBPK model with 12 compartments. Excretion of the compound is from feces via gut, urine via kidney and breath via lung, respectively.

**Figure 4 ijerph-20-03473-f004:**
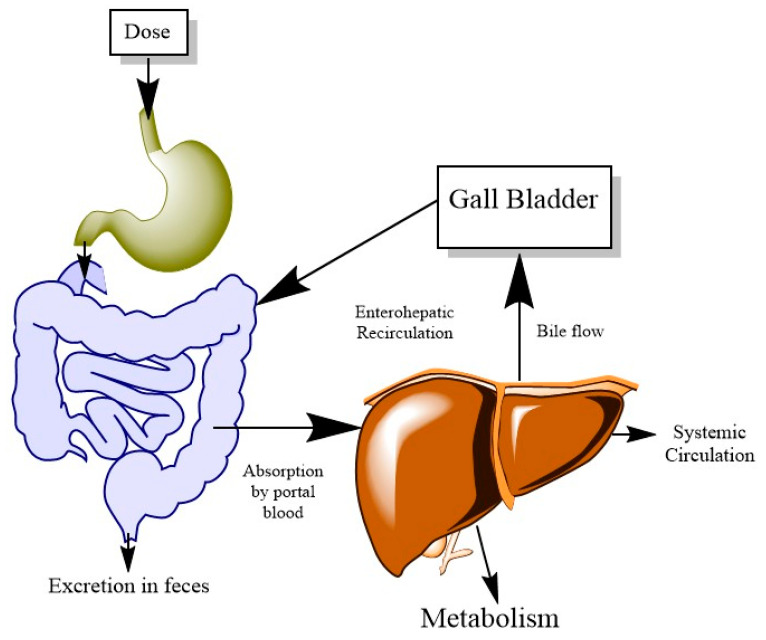
Oral absorption of xenobiotic followed by enterohepatic recirculation (EHR). EHR occurs by biliary excretion followed by intestinal reabsorption of xenobiotic along with hepatic conjugation and intestinal deconjugation.

**Figure 5 ijerph-20-03473-f005:**
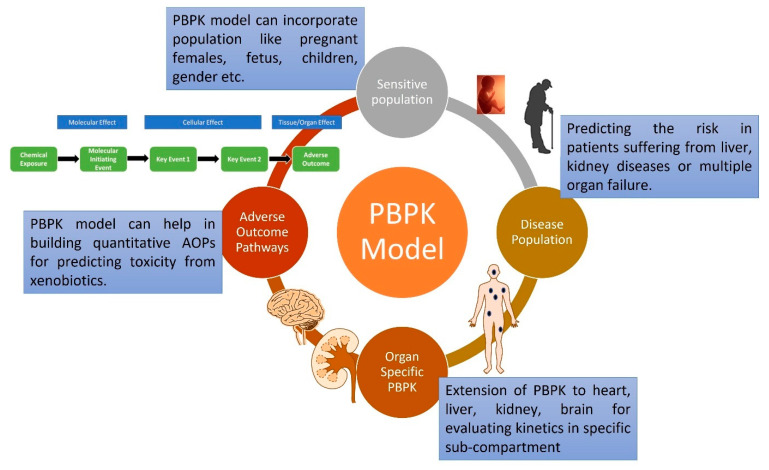
Applications of PBPK models in different fields (sensitive populations, diseased populations, organ-specific and adverse outcomes) for therapeutics and toxicological risk assessment.

**Figure 6 ijerph-20-03473-f006:**
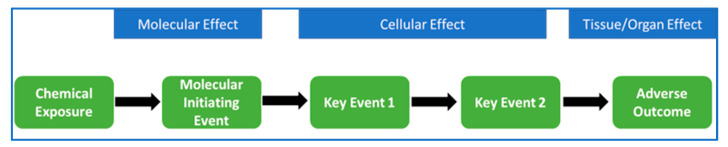
AOP scheme.

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
