# Peer review of "The Role of “Physiologically Based Pharmacokinetic Model (PBPK)” New Approach Methodology (NAM) in Pharmaceuticals and Environmental Chemical Risk Assessment"

_ijerph, 2023, doi:10.3390/ijerph20043473_

Round 1

Reviewer 1 Report

The manuscript entitled "The role of "Physiologically Based Pharmacokinetic Model (PBPK)" New Approach Methodology (NAMs) in Pharmaceuticals and Environmental Chemical Risk Assessment" presents the current state of the art in PBPK models and their use in the field of environmental and health risk assessment. The topic is very important in the field of risk assessment; in particular, the authors have attempted to highlight the importance and use of the PBPK model, as well as the various approaches to integrating the PBPK model into AOPs. However, there is room for improvement, so I have some comments/suggestions that the authors could consider.

Comments:

In Section 3.3, the authors have attempted to show the application of the PBPK model in the field of IATA and AOP.  The authors have discussed how PBPK can help in decision-making for risk assessment. However, I see that there are still many points that need to be covered in this section, e.g.: The ultimate goal of AOP is often to predict damage at the organism/population level. The author could discuss how PBPK can be very helpful by creating a virtual population by considering variability within the population or even considering special populations or diseased populations to predict the risk that is very difficult to determine from the in vitro study alone. The question of how risk assessment can be facilitated by integrating the PBPK model into the AOP is also not entirely clear. For example, what factor can be included in the PBPK models that will provide better insight into the kinetic changes in dynamics that are often difficult to assess in in vitro cells? Since in vitro kinetics may be very different from in vivo kinetics, the author should discuss the challenges and opportunities that may arise. Finally, I think uncertainty and variability play a critical role in integrating PBPK into AOPs; it would be nice to discuss this.

Minor remarks:

The author stated, "All equations can be converted to concentration by dividing the amount by the volume of a given tissue." In the case of the PBPK model, this is indeed true, as one considers the total blood volume and the total body volume in which the drug can be distributed with a mass balance by considering the "rest of the body." But I think if you use an empirical model with only a few compartments, which was given as an example by the author, you often have to consider the volume of distribution, which is often fitted.

Equation 5: I missed the information about the parameters and variables, this can confuse the reader a lot.

Equation 9: The equation is not very well described, for example, it needs to be explained why the value of 60 is multiplied or divided by 10 to the 9th power.

The author should also emphasize how metabolic kinetics data have often been given and how they can be scaled, e.g. in terms of the metabolic rate per cell number.

Line 424, the citation is missing

“Mechanistical tools” -> Mechanistic tools; I guess there is no such word “mechanistical”

Author Response

We would like to sincerely thanks the reviewers who have spent their valuable time in providing comments for improving the review. We have tried our best to include all the comments in the revised manuscript.

Reviewer 1:

The manuscript entitled "The role of "Physiologically Based Pharmacokinetic Model (PBPK)" New Approach Methodology (NAMs) in Pharmaceuticals and Environmental Chemical Risk Assessment" presents the current state of the art in PBPK models and their use in the field of environmental and health risk assessment. The topic is very important in the field of risk assessment; in particular, the authors have attempted to highlight the importance and use of the PBPK model, as well as the various approaches to integrating the PBPK model into AOPs. However, there is room for improvement, so I have some comments/suggestions that the authors could consider.

Comments:

In Section 3.3, the authors have attempted to show the application of the PBPK model in the field of IATA and AOP.  The authors have discussed how PBPK can help in decision-making for risk assessment. However, I see that there are still many points that need to be covered in this section, e.g.: The ultimate goal of AOP is often to predict damage at the organism/population level. The author could discuss how PBPK can be very helpful by creating a virtual population by considering variability within the population or even considering special populations or diseased populations to predict the risk that is very difficult to determine from the in vitro study alone. The question of how risk assessment can be facilitated by integrating the PBPK model into the AOP is also not entirely clear. For example, what factor can be included in the PBPK models that will provide better insight into the kinetic changes in dynamics that are often difficult to assess in in vitro cells? Since in vitro kinetics may be very different from in vivo kinetics, the author should discuss the challenges and opportunities that may arise. Finally, I think uncertainty and variability play a critical role in integrating PBPK into AOPs; it would be nice to discuss this.

Answer: Thanks for the comment. We have included paragraph in the section 3.3 where we focused on how PBPK can play a role in building AOPs and also the role of uncertainty and variability.

Minor remarks:

The author stated, "All equations can be converted to concentration by dividing the amount by the volume of a given tissue." In the case of the PBPK model, this is indeed true, as one considers the total blood volume and the total body volume in which the drug can be distributed with a mass balance by considering the "rest of the body." But I think if you use an empirical model with only a few compartments, which was given as an example by the author, you often have to consider the volume of distribution, which is often fitted.

Answer: I agree with the reviewer. Indeed, in case of 1-2 compartmental model, the concentration can be obtained by considering apparent volume of distribution whereas in multi-compartment model, we consider the volume of a particular tissue. The modification has been done in the manuscript.

Equation 5: I missed the information about the parameters and variables, this can confuse the reader a lot.

Answer: The information about the equation 5 has been provided in the text.

Equation 9: The equation is not very well described, for example, it needs to be explained why the value of 60 is multiplied or divided by 10 to the 9th power.

Answer: The information has been provided in the text about conversion of units.

The author should also emphasize how metabolic kinetics data have often been given and how they can be scaled, e.g., in terms of the metabolic rate per cell number.

Answer: The information for representation of metabolic kinetics data and scaling has been added in the section “metabolism with IVIVE”.

Line 424, the citation is missing

Answer: The citation has been added.

“Mechanistical tools” -> Mechanistic tools; I guess there is no such word “mechanistical”

Answer: Modification has been done to “mechanistic”.

Reviewer 2 Report

This is an area really required more extensive literature and authors tried to present it good way. After reading this paper I would like to learn few information which are written as below:

1-What are the value-added information provided in this paper? It appears all the basic information already available in literature.

2- What if it would be a systematic review or more meta-analysis type of perennation of different drugs, chemicals and associated models and populations available in literature.

3- It is also suggested to present the literature gap analysis and guidelines for PBPK related quality.

Author Response

We want to thank the reviewer for their useful comments for improving the manuscript.

This is an area really required more extensive literature and authors tried to present it good way. After reading this paper I would like to learn few information which are written as below:

1-What are the value-added information provided in this paper? It appears all the basic information already available in literature.

Answer: I agree with the reviewer that there is some basic information already available in literature. However, we have not limited ourselves to the basic information rather we tried to introduce some novel topics which are quite new in PBPK itself. For instance, we have introduced some advanced concepts like utilizing PBPK models in qAOP development. As per our knowledge currently there are few selected papers related to this. We struggled to include some citations for this aspect but there is a future scope for this research area. Also, we have introduced machine learning in PBPK which is still in its infancy stage. Such topics can give idea to the researchers about the dimensions in which PBPK is still evolving and a lot need to be done over coming years. Generally, the basic information in literature about PBPK is quite complex for non-computational readers to understand especially from applicability and modeling point of view. The paper has been written in such a way that non-experienced scientist can understand it very well and for expert’s scientist we have also introduced some advanced PBPK. The organ-specific PBPK have been mentioned like brain-specific PBPK, and how we can model them. This information is highly helpful for expert scientist who want to advance their knowledge and expertise. We also tried to cover some regulatory aspect and limitations of existing PBPK model so that better guidelines can be built.

2- What if it would be a systematic review or more meta-analysis type of perennation of different drugs, chemicals and associated models and populations available in literature.

Answer: We believe that systemic review could be a good area of work where we have sufficient literature  to review and build some opinion or understand trends or even research gaps. However, most of PBPK modelling literatures are about technical aspects of model of development and validation. Application of PBPK models as NAM or in IATA is almost nonexistence. However, with applied research works getting published with PBPK models, this is something which can be published in future. The basic idea behind this review is not to cover all the papers related PBPK models but rather point out the recent techniques and guide the readers about where PBPK are heading. We have used expertise of our group as we are working on PBPK from last 15 years or so to provide direction to readers about the future of PBPK and its role in the development of adverse outcome pathways. This manuscript is also a discussion documents in a large EU project European Partnership for the Assessment of Risks from Chemicals (PARC), where development of NAM and inclusion of PBPK models in IATA are being discussed.

3- It is also suggested to present the literature gap analysis and guidelines for PBPK related quality.

Answer: Thanks for a very good and constructive comment. We have tried to address this comment and introduced a new paragraph in section 5 on literature gap analysis and guidelines for PBPK by agencies like EMA, and USFDA.

Round 2

Reviewer 2 Report

Thank you for the revision.